# Transcriptomic Analysis of the Dehydration Rate of Mature Rice (*Oryza sativa*) Seeds

**DOI:** 10.3390/ijms241411527

**Published:** 2023-07-16

**Authors:** Zhongqi Liu, Jinxin Gui, Yuntao Yan, Haiqing Zhang, Jiwai He

**Affiliations:** College of Agronomy, Hunan Agricultural University, Changsha 420128, China; liuzhongqi886@163.com (Z.L.); smileyyt1314@163.com (Y.Y.)

**Keywords:** rice seeds, dehydration rate, RNA-seq, DEGs

## Abstract

In this study, a transcriptomic analysis of the dehydration rate of mature rice seeds was conducted to explore candidate genes related to the dehydration rate and provide a theoretical basis for breeding and utilization. We selected two rice cultivars for testing (Baghlani Nangarhar, an extremely rapid dehydration genotype, and Saturn, a slow dehydration genotype) based on the results determined by previous studies conducted on the screening of 165 germplasm materials for dehydration rate phenotypes. A rapid dehydration experiment performed on these two types of seeds was conducted. Four comparative groups were set up under control and dehydration conditions. The differentially expressed genes (DEGs) were quantified via transcriptome sequencing and real-time quantitative PCR (RT-qPCR). GO (Gene ontology) and KEGG(Kyoto Encyclopedia of Genes and Genomes) analyses were also conducted. In Baghlani Nangarhar, 53 DEGs were screened, of which 33 were up-regulated and 20 were down-regulated. In Saturn, 25 DEGs were screened, of which 19 were up-regulated and 6 were down-regulated. The results of the GO analysis show that the sites of action of the differentially expressed genes enriched in the rapid dehydration modes are concentrated in the cytoplasm, internal components of the membrane, and nucleosomes. They play regulatory roles in the processes of catalysis, binding, translocation, transcription, protein folding, degradation, and replication. They are also involved in adaptive responses to adverse external environments, such as reactive oxygen species and high temperature. The KEGG analysis showed that protein processing in the endoplasmic reticulum, amino acid biosynthesis, and oxidative phosphorylation were the main metabolic pathways that were enriched. The key differentially expressed genes and the most important metabolic pathways identified in the rapidly and slowly dehydrated genotypes were protein processing in the endoplasmic reticulum and oxidative phosphorylation metabolism. They were presumed to have important regulatory roles in the mechanisms of stress/defense, energy metabolism, protein synthesis/folding, and signal transduction during the dehydration and drying of mature seeds. The results of this study can potentially provide valuable information for further research on the genes and metabolic pathways related to the dehydration rate of mature rice seeds, and provide theoretical guidance for the selection and breeding of new rice germplasm that can be rapidly dehydrated at the mature stage.

## 1. Introduction

Rice (*Oryza sativa*) is one of the most important food crops in China. Rice produces orthodox seeds, which can be dried to achieve a low moisture content and stored at low temperatures to prolong their viability. A high moisture content of orthodox seeds during maturity and harvesting decreases their viability and longevity properties. The rate of dehydration directly affects the safe harvesting and rapid drying of the seeds. Therefore, it is especially important to select varieties with a low moisture content and rapid dehydration rate at maturity to ensure high seed quality and reduce production costs. Some progress has been made in the research on different types of seed dehydration tolerance and dehydration rate measures, e.g., in *Arabidopsis* [1], *Maize* [2,3], *Wheat* [4], *Coffee* [5], *Trichoderma* [6], and sand mustard seeds [7]. Some QTLs (Quantitative trait locus) and genes related to the dehydration rate and dehydration tolerance of *maize* seeds have been obtained from a molecular biology perspective [8,9,10,11,12,13]. However, further studies are required to understand how to activate genes associated with seed dehydration tolerance, how dehydration genes respond to dehydration stress signals, and how gene co-expression network mechanisms work.

Transcriptomic analysis is currently an important approach to studying gene function. It can help to reveal the functions of various cellular tissue components, specific biological processes, and the molecular functions exercised. It is now widely used in the study of rice, *maize*, oilseed rape, and other crops. Transcriptomics has made some progress in plant dehydration tolerance studies. In one study, Costa et al. used transcriptomics to dynamically examine the effect of abscisic acid (ABA) on the dehydration tolerance rate of *Arabidopsis* seed germination, and the results showed that germination was more likely to be induced in mature *Arabidopsis* seeds [14]. In another study, Cecília et al. obtained further microarray data from the time series for *Arabidopsis* seeds, a network of gene co-expression was established, including two regions: early response (ER) and late response (LR) [15]. A further study of *Arabidopsis* seeds obtained two specific transcriptional regulatory networks of dehydration tolerance (TFsSeed-subNetDT1 and TFsSeed-subNetDT2) with storage compounds and cytoprotective mechanisms [16]. This regulation of key genes for dehydration tolerance is manifold under extreme dehydration drying conditions [17]. Transcription factors related to dehydration tolerance, such as CaHSFA9, CaDREB2G, CaANAC029, CaPLATZ, and CaDOG-like, were obtained through transcriptomics during a study of intermediate coffee seeds [18]. In another study, dehydration-associated gene co-expression clusters were observed to be functionally enriched in the seed development process, and these genes were involved in various pathways, such as programmed cell death inhibition and the activation of ABA signaling in a trans-metabolic network [19]. The use of transcriptomics to study cork oak seed dehydration sensitivity yielded 2219 DEGs [20]. These differentially expressed genes are mainly related to hormone biosynthesis (IAA, ABA) and signaling (ZEP, PYR, YUC, ERF1B, ABI5, etc.), stress response proteins (HSP70, LEA D-29, etc.), and phospholipase D (PLD1) [20]. Seed desiccation sensitivity may be determined by these genes and their interactions. Transcriptional studies conducted on the effect of exogenous ABA on the rate of dehydration in *maize* cob position leaves and ears were used to obtain 73 differentially expressed genes related to water metabolism, and these genes were hypothesized to be exogenous ABA-regulated downstream genes involving multiple transcription factors [21]. A combined multi-omics analysis was used to identify 143 genes significantly associated with *maize* kernel water content and kernel dehydration rate, which are closely related to biological processes such as starch and fatty acid biosynthesis, cold stress, and salt stress [22].

However, few studies exist in the literature on the dehydration tolerance of rice seeds following their harvest, and there is no report of any genetic studies on the dehydration rate of mature rice seeds. In this study, we selected Baghlani Nangarhar from Afghanistan, an extremely rapidly dehydrated genotype, and Saturn from the United States, a slowly dehydrated genotype, from the core germplasm of rice from 82 countries and regions, and we verified the quantitative expression of the screened differentially expressed genes by applying transcriptome sequencing and real-time quantitative PCR methods during the seed dehydration process at the rice maturity level. The aims of the study were to elucidate the molecular mechanisms involved in the seed dehydration rate at rice maturity, to uncover the expression patterns and metabolic pathways of key differentially expressed genes affecting the dehydration rate, and to provide a new theoretical basis for the creation of new varieties of mature rice that can be dried in sun-free conditions.

## 2. Results

### 2.1. Dehydration Rate Performance of Rice Core Seed in Relation to Rapid and Slow Dehydration Genotypes

In this study, we determined that the moisture content of rice seeds during physiological maturity was generally in the range of 20–30% and gradually decreased with increasing maturity. The average initial moisture content at harvest was 19.0% for the temperate japonica-type seed, Baghlani Nangarhar, and 18.3% for the intermediate-type seed, Saturn, with little difference in the moisture content levels between them. The rate of dehydration decreased as the dehydration time increased, and the maximum moisture reduction outcome was achieved after 4 h of dehydration, with the difference reaching a highly significant level and reducing after 12 h (Figure 1). This result indicates that the dehydration rate of Baghlani Nangarhar is significantly quicker than that of Saturn.

### 2.2. Transcriptome Sequencing Data Statistics

Samples collected from Baghlani Nangarhar that were dehydrated under control versus dehydrated conditions were labeled “CNSF5” and “NSF5”, and samples collected from Saturn that were dehydrated under control versus dehydrated conditions were labeled “CNSF75” and “NSF75”. A total of four comparison groups were constructed using these samples. Rapidly dehydrated Baghlani Nangarhar seeds and slowly dehydrated Saturn seeds at physiological maturity were selected for transcription sequencing, and three replicates of twelve samples were set up for control groups CNSF5-1, CNSF5-2, CNSF5-3, CNSF75-1, CNSF75-2, and CNSF75-3, and treatment groups NSF5-1, NSF5-2 NSF5-3, NSF75-1, NSF75-2, and NSF75-3. Good-quality RNA results were obtained from the 12 samples (Appendix A). From Appendix A, it can be observed that the concentration levels of RNA in all 12 samples ranged from 418 to 1395 ng/µL, OD28S/18S ranged from 1.50 to 2.20, OD260/280 ranged from 1.88 to 2.19, and the RIN values ranged from 7.60 to 9.60. There were three main reasons for this: (1) when we extracted the RNA, we added DNase I for the DNA digestion. (2) The principle of transcription library construction technology is to use oligodT magnetic beads to specifically enrich mRNA with polyA. DNA does not have this feature; therefore, it will not bind. If rRNA removal technology is used to build a library (such as Lnc RNA products), there is also a step of DNA digestion included in the technical process. (3) Our second-generation RNA library construction and sequencing did not require an OD value; however, it mainly focused on the RIN value, 28S/18S, and total amount. In addition, the highest absorption peak of DNA and RNA was at 260 nm; therefore, the OD value could not distinguish between DNA and RNA. Thus, the nanodrop detection method produced relatively inaccurate results for the quantification of both DNA and RNA. These results indicate that the RNAs of all samples used in this experiment were intact, free of impurities, and of good quality for sequencing library construction, and were ready to use for the sequencing experiments. The GC content distribution results show that there was no separation of AT/GC in each sample, and the GC content was 51.12% and above in all samples (Appendix A). The base quality value (Q30) of each sample was greater than 93.73, the average matching rate was 93.91% when compared with the sequence of the rice reference genome (Nihon Haru), and the average matching rate was 81.94% for the unique sequence comparison (Appendix A). The abovementioned results indicate that the library construction quality of the transcriptional sequencing method is good. The final 12 samples exposed to whole-transcriptome high-throughput sequencing each presented an average of 1.18 Gb of clean sequences, and a total of 5633 non-repeat differentially expressed genes were detected (Appendix A).

### 2.3. Identification of Differentially Expressed Genes

A total of four comparison groups were constructed: CNSF5 vs. NSF5, CNSF5 vs. CNSF75, NSF5 vs. NSF75, and CNSF75 vs. NSF75. As shown in Figure 2A, 53, 4307, 4736, and 25 DEGs were identified in the four comparative combinations CNSF5 vs. NSF5, CNSF5 vs. CNSF75, NSF5 vs. NSF75, and CNSF75 vs. NSF75, respectively. This indicates that the significant differences in gene expression levels not only occurred between different treatments, but also between different varieties. In the comparison group CNSF5 vs. NSF5, 33 genes with up-regulated expressions and 20 genes with down-regulated expressions were identified; in CNSF5 vs. CNSF75, 1598 genes with up-regulated expressions and 2709 genes with down-regulated expressions were identified; in the comparison group NSF5 vs. NSF75, 1941 genes with up-regulated expressions and 2795 genes with down-regulated expressions were identified; and in the comparison group CNSF75 vs. NSF75, 19 genes with up-regulated expressions and 6 genes with down-regulated expressions were identified (Figure 2A). Overall, NSF5 showed more up-regulated expressions of DEGs compared to CNSF5 under rapid dehydration conditions. In addition, more DEGs were present in the rapidly dehydrated genotype compared to the dehydrated genotype; this indicates that the the intrinsic properties of the two rice varieties had a greater effect on the dehydrated Baghlani Nangarhar genotype than the dehydrated Saturn genotype.

A total of 5633 non-duplicate DEGs were included in the four comparison groups (Figure 2B). Among them, CNSF5 vs. NSF5 occupied 0.39% (22/5633), CNSF5 vs. CNSF75 occupied 15.21% (857/5633), NSF5 vs. NSF75 occupied 22.81% (1285/5633), and CNSF75 vs. NSF75 occupied 0.089% (5/5633) (Figure 2B). Since the CNSF5 and CNSF75 samples in CNSF5 vs. CNSF75 were not treated with the rapid dehydration method, the differentially expressed genes appearing within this combination were treated as DEGs between these two genotypes (4307), and the differentially expressed genes in the remaining three comparison groups were treated as DEGs related to the dehydration rate (1326).

### 2.4. GO Enrichment Analysis of Differentially Expressed Genes

All DEGs were annotated with gene functions and classified into three major categories, i.e., cellular component, molecular function, and biological process, via GO enrichment analysis. CNSF5 vs. NSF5 had 53 DEGs, annotated into 38 entries in molecular functions, 28 entries in cellular components, and 46 entries in biological processes, for a total of 112 entries. The largest number of differentially expressed genes in the molecular function category related to unfolded protein binding (GO:0051082, 7 entries); the largest number of DEGs in the cellular fraction category related to cytoplasm (GO:0005737, 10 entries); and the largest number of DEGs in the biological process category related to pro-folded protein (GO:0005737, 10 entries). The entry with the highest number of DEGs in the biological processes category was protein folding (GO:0006457, 6 entries). From the GO enrichment results, it can be observed that the molecular functions performed by CNSF5 vs. NSF5 are significantly enriched in (1) unfolded protein binding (GO:0051082, 7, Q-value = 7.56 × 10^−8^), (2) protein self-association (GO:0043621, 4, Q-value = 8.95 × 10^−6^); and (3) heat-shock protein binding (GO:0031072, 2, Q-value = 0.0177) processes; the significantly enriched entries in cellular fractions are cytoplasm (GO:0005737, 10, Q-value = 7.81 × 10^−3^). The entries significantly enriched in biological processes are (1) protein folding (GO:0006457, 6, Q-value = 3 × 10^−7^), (2) response to heat (GO:0009408, 5, Q-value = 5.08 × 10^−7^), (3) protein complex oligomerization (GO:0051259, 4, Q-value = 3.40 × 10^−7^), (4) response to hydrogen peroxide (GO:0042542, 4, Q-value = 6.76 × 10^−7^), (5) response to reactive oxygen species (GO:0000302, 4, Q-value = 1.03 × 10^−6^), and (6) response to salt stress (GO:0009651, 4, Q-value = 1.02 × 10^−4^) (Figure 3A–C, Appendix A).

CNSF5 vs. CNSF75 had a total of 4307 DEGs, annotated into 828 entries in molecular functions, 297 entries in cellular components, and 1109 entries in biological processes, for a total of 2234 entries. Among them, ATP binding (GO:0005524, 385) was the entry with the highest number of DEGs in the molecular function category; the internal component of the membrane (GO:00160, 813) was the entry with the highest number of DEGs in the cellular component category; the regulation of DNA-templated transcription (GO:0006355, 150) genes included entries with the highest number of DEGs involved in biological processes. As shown by the GO enrichment results, the significantly enriched entries in the molecular functions of CNSF5 vs. CNSF75 were mainly (1) ADP binding (GO:0043531, 96, Q-value = 4.56 × 10^−5^), (2) protein heterodimerization activity (GO:0046982, 31, Q-value = 0.0089), (3) UDP-glycosyltransferase activity (GO:0008194, 40, Q-value = 0.0139), and (4) ligand-gated ion channel activity (GO:0015276, 11, Q-value = 0.0209). The significantly enriched entries for cellular components were nucleosome (GO:0000786, 31, Q-value = 1.75 × 10^−8^), integral component of membrane (GO:0016021, 813, Q-value = 4.13 × 10^−6^), and plasma membrane (GO:0005886, 260, Q-value = 0.0173); the entries that were significantly enriched in biological processes were (1) nucleosome assembly (GO:0006334, 19, Q-value = 0.0032), (2) defense response (GO:0006952, 92, Q-value = 2.61 × 10^−4^), and (3) transmembrane transport (GO:0055085, 70, Q-value = 3.39 × 10^−4^) (Figure 3D–F).

NSF5 vs. NSF75 had a total of 4736 DEGs, annotated into 858 entries in molecular functions, 331 entries in cellular components, and 1176 entries in biological processes, for a total of 2365 entries. The largest number of differentially expressed genes in these molecular functions were enriched in ATP binding (GO:0005524, 419); the largest number of DEGs in cellular components were enriched in the internal component of the membrane (GO:0016021, 872); and the largest number of DEGs in biological processes were enriched in ATP binding (GO:0005524, 419). The entry with the highest number of DEGs enriched in biological processes was the regulation of DNA-templated transcription (GO:0006355, 174). From the GO enrichment results, we can observe that (1) ADP binding (GO:0043531, 99, Q-value = 4.99 × 10^−4^), (2) UDP-glycosyltransferase activity (GO:0008194, 45, Q-value = 0.00192), (3) protein heterodimerization activity (GO:0046982, 34, Q-value = 0.00192), (4) ligand-gated ion channel activity (GO:0015276, 12, Q-value = 0.00848), (5) nutrient reservoir activity (GO:0045735, 26, Q-value = 0.0186), and (6) quercetin 3-O-glucosyltransferase activity (GO:0080043, 16, Q-value = 0.0258) were the most significantly enriched entries in the molecular functions of the NSF5 vs. NSF75 comparison group. Nucleosome (GO:0000786, 27 entries, Q-value = 1.14 × 10^−4^), an integral component of the membrane (GO:0016021, 872 entries, Q-value = 4.59 × 10^−4^), was the entry significantly enriched in cellular components, while (1) nucleosome assembly (GO:0006334, 17, Q-value = 0.123), (2) transmembrane transport (GO:0055085, 76, Q-value = 0.123), (3) cell proliferation (GO:0008283, 7, Q-value = 0.123), (4) plant-type primary cell-wall biogenesis (GO:0009833, 12, Q-value = 0.123), (5) cellulose biosynthetic process (GO:0030244, 13, Q-value = 0.152), and (6) response to water deprivation (GO:0009414, 22, Q-value = 0.720) (Figure 3G–I) were also evaluated.

CNSF75 vs. NSF75 had 25 DEGs, annotated as having 22 entries in the molecular functions category, 20 entries in the cellular components category, and 23 entries in the biological processes category, for a total of 65 entries. Among them, unfolded protein binding (GO:0051082, 10) presented the largest number of DEGs concerning molecular function; cytoplasm (GO:0005737, 10) presented the largest number of DEGs concerning cellular fraction; and protein-folding (GO:0006457, 10) entries presented the highest number of DEGs concerning biological processes. From the GO enrichment results, it can be observed that the molecular functions performed by CNSF75 vs. NSF75 are significantly enriched for (1) unfolded protein binding (GO:0051082, 10, Q-value = 1.9 × 10^−14^), (2) protein self-association (GO:0043621, 7, Q-value = 5.2 × 10^−13^), (3) DNA-binding transcription factor activity (GO:0003700, 5, Q-value = 0.0130), (4) RNA polymerase II proximal promoter sequence-specific DNA-binding molecule (GO:0000978, 2, Q-value = 0.0229), (5) triose-phosphate transmembrane transporter activity (GO:0071917, 1, Q-value = 0.0229), and (6) phosphoglycerate transmembrane transporter activity (GO:0015120, 1, Q-value = 0.02361). Cytoplasm (GO:0005737, 10, Q-value = 1.22 × 10^−4^), cell surface (GO:0009986, 2, Q-value = 1.22 × 10^−4^), protein-containing complex (GO:0032991, 2, Q-value = 1.22 × 10^−4^), and the perinuclear region of cytoplasm (GO:0048471, 2, Q-value = 5.51 × 10^−4^) are significantly enriched entries in cellular fractions; protein folding (GO:0006457, 10, Q-value = 3.36 × 10^−16^), protein complex oligomerization (GO:0051259, 7, Q-value = 8.49 × 10^−16^), response to hydrogen peroxide (GO:0042542, 7, Q-value = 8.60 × 10^−15^), response to reactive oxygen species (GO:0000302, 7, 1.83 × 10^−14^), response to heat (GO: 0009408, 8, Q-value = 2.20 × 10^−14^), response to salt stress (GO:0009651, 7, Q-value = 7.88 × 10^−11^), and cellular response to heat (GO:0034605, 4, Q-value = 1.81 × 10^−7^) are significantly enriched entries involved in biological processes (Figure 3J–L, Appendix A).

### 2.5. KEGG Enrichment Analysis of Differentially Expressed Genes

In the four comparison groups, CNSF5 vs. NSF5, CNSF5 vs. CNSF75, NSF5 vs. NSF75, and CNSF75 vs. NSF75, 53 DEGs were allocated to 9 KEGG pathways, 4307 DEGs to 120 KEGG pathways, 4736 DEGs to 123 KEGG pathways, and 25 DEGs to 4 KEGG pathways, respectively. The top-20 most significantly different genes in each comparison group were selected for bubble plots (Figure 4A–D).

The metabolic pathway with the highest number and most significant enrichment of KEGG-enriched DEGs in comparison group CNSF5 vs. NSF5 was protein processing in the endoplasmic reticulum (osa4141, 6 entries, Q-value = 3.17 × 10^−5^), with the following associated genes: LOC_Os01g04340 (HSP16.6, Log2FC: 1.51), LOC_Os03g16040 (HSP17.7, Log2FC: 1.22), LOC_Os03g16020 (HSP17.4, Log2FC: 1.33), LOC_Os03g16030 (HSP18.1, Log2FC: 1.94), LOC_Os04g01740 (HSP82, Log2FC: 1.54), and LOC_Os03g53340 (HSF11, Log2FC: 1.16). The expressions of these differentially expressed genes were significantly up-regulated in NSF5. The expression of gene Os08g0400200 in the lysine degradation (osa00310) pathway was significantly down-regulated in NSF5 (Log2FC: −1.15); gene LOC107279585 in the oxidative phosphorylation pathway (osa00190) was significantly up-regulated in NSF5 (Log2FC: 1.49). In addition, chitinase 2-like (LOC_Os05g33130) was involved in both amino sugar and nucleotide sugar metabolism (osa00520) processes and the plant MAPK signaling pathway (osa04016), and its expression level was significantly up-regulated (Log2FC: 1.40). It was hypothesized that the LOC_Os05g33130 gene possesses catalytic and transport functions. The gene Os03g0738200 participates in the messenger RNA surveillance pathway (osa03015), and its expression was significantly down-regulated (Log2FC: −1.08); the gene Os05g0163250 is involved in the splicing pathway (osa03040), and its expression was significantly down-regulated (Log2FC: −1.03) (Figure 4A, Appendix A). The KEGG metabolic pathway of other genes is unknown.

The metabolic pathways with the highest number of KEGG-enriched DEGs in the comparison group CNSF5 vs. CNSF75 were biosynthesis of amino acids (osa1230, 41 entries), phenylpropanoid biosynthesis (osa940, 38 entries), plant–pathogen interaction (osa4626, 37 entries), and carbon metabolism (osa1200, 36 entries). From the perspective of significant enrichment, the metabolic pathways involved in this process were alanine, aspartate, and glutamate metabolism (osa00250, Q-value = 0.120), tyrosine metabolism (osa00350, Q-value = 0.120), and sphingolipid metabolism (osa600, Q-value = 0.120) (Figure 4B).

In the comparison group NSF5 vs. NSF75, in terms of the number of DEGs enriched by KEGG, the results show that the most enriched metabolic pathway was ribosome (osa3010, 64), followed by biosynthesis of amino acids (osa1230, 40), plant–pathogen interaction (osa4626, 39), carbon metabolism (osa1200, 39), phenylpropanoid biosynthesis (osa940, 38), and plant hormone signal transduction (osa4075, 32). The metabolic pathways of significant enrichment were galactose metabolism (osa52, Q-value = 0.0970), homologous recombination (osa3440, Q-value = 0.0970), ribosomes (osa3010, Q-value = 0.0970), and fatty acid elongation (osa62, Q-value = 0.0988) (Figure 4C).

The metabolic pathway with the highest number and most significant enrichment of DEGs in the comparison group CNSF75 vs. NSF75 was protein processing in the endoplasmic reticulum (osa4141, 9, Q-value = 1.71 × 10^−10^), followed by plant–pathogen interaction (osa4626, 2). ATP synthase subunit 9 (LOC107279585) was significantly up-regulated in NSF75 (Log2FC: 1.16, Q-value = 0.05) and NSF5 (Log2FC: 1.49, Q-value = 7.280 × 10^−11^). LOC_Os04g01740 (HSP82, chr4:483216-486030) was significantly up-regulated in NSF75 (Log2FC:1.74, Q-value = 2.7 × 10^−11^) and NSF5 (Log2FC:1.54, Q-value = 0.0163); LOC_Os03g16020 (HSP17.4, chr3: 8833703-8834481) was significantly up-regulated in NSF75 (Log2FC:1.23, Q-value = 0.000754); and LOC_Os03g16030 (HSP18.1, chr3:8834792-8835644) was significantly up-regulated in NSF75 (Log2FC:2.36, Q-value = 0.00484) (Figure 4D, Appendix A) [23]. These heat-shock protein-related genes are involved in protein-processing activity in the endoplasmic reticulum and ATP synthesis and metabolism pathways. The up-regulation of their expression may be a timely positive response during dehydration and drying processes, adapting to the environment, and protecting cells from damage.

### 2.6. Analysis of Dehydration-Related Candidate Genes in Transcriptome

#### 2.6.1. Screening of Candidate Genes in Rapid and Slow Dehydration Materials

There were 10 significantly differentially expressed genes (padj < 0.01, │Log2FC│ ≥ 1) present in the rapidly dehydrated genotype (Baghlani Nangarhar) and the slowly dehydrated genotype (Saturn) (Figure 5A), of which 7 differed (Fragments Per Kilobase of exon model per Million mapped fragments) by twice (Log2FC│ ≥ 1)) the amount or more, and all of these DEGs were consistently up-regulated in expression (Table 1), suggesting that they may play an important role in the seed dehydration process. These genes are mainly involved in the protein-processing metabolic pathway in the endoplasmic reticulum and the oxidative phosphorylation pathway (Figure 5B). Heat-stimulated transcription factor A6A, heat-stimulated transcription factor A2A, ethylene-responsive transcription factor ABR1, ATP synthesis subunit 9, small-molecule heat-stimulated protein HSP20, and heat-stimulated protein HSP90 were also involved in this process.

Further analysis showed that the differentially expressed gene *Os06g0722450* [Log2FC: 8.28] was up-regulated, *Os06g0513943* [Log2FC: 1.82] was up-regulated, and *LOC112938716* [Log2FC: −2.42] was down-regulated, and the three genes(*Os06g0722450*, *Os06g0513943*, *LOC112938716*) were only significantly expressed in the rapid dehydration genotype Baghlani Nangarhar. The differentially expressed gene *Os06g0159900* [Log2FC: 1.03], which was significantly expressed only in the slow dehydration genotype, was up-regulated, and *LOC9266706* [Log2FC: −8.25] was down-regulated. Among them, *Os06g0722450* was an expression protein with an unknown function, which was not expressed in the initial state of the rapid dehydration genotype, and was activated and up-regulated during the dehydration process. Its target mRNA is *novel-osa-miR194-3p* [23], which was presumed to be a new gene related to the dehydration rate. *Os06g0513943* was up-regulated in the rapid dehydration genotype; its main functions are to maintain the integrity of the membrane, reduce physical damage to the seeds, and play a self-defense role [24]. *LOC112938716* was down-regulated in the rapid dehydration genotype, indicating that its expression was inhibited during dehydration. *Os06g0159900* belongs to the U-box domain protein [25,26,27], which was highly up-regulated in the slow dehydration genotype. Its molecular functions include ubiquitin protein transferase and kinase activities, which can regulate the internal and external pressure balance levels of seed cells. *LOC9266706* is an expression protein with an unknown function, which was down-regulated in a large amount of the slow dehydration genotype, and was expressed in the initial state. Following dehydration and drying stages, the expression level of *LOC9266706* was close to zero. It was hypothesized that its expression was inhibited during the seed dehydration process. However, the gene was up-regulated in the rapid dehydration genotype, and it was predicted to be a new gene *novel-osa-miR116-5p* through targeted miRNA information.

The GO enrichment analysis of the comparison group NSF5 vs. NSF75 yielded 45 genes related to water metabolism, including the gene PIP1;1 (LOC_Os02g44630) (Appendix A), which was induced to be up-regulated in both the rapidly dehydrated genotype and the slow dehydrated genotype. The expression of the gene PIP1;1 (LOC_Os02g44630) was 1.67 times higher in the rapidly dehydrated genotype than in the slowly dehydrated genotype. It was assumed that this gene had an important role in the regulation of water transport channel activity during seed maturation and drying processes. This result is consistent with the results obtained by Valérie [12], who identified the up-regulated expression of water channel proteins (genes) (OsPIP1, OsPIP2) associated with seed dehydration activity using the QTL localization method to study the relationship between seed dehydration and ABA content in maize seeds.

#### 2.6.2. Validation of Transcriptome Sequencing Genes via RT-qPCR

Based on the results obtained for the previous screening of 165 germplasm materials for dehydration rate phenotypes, two extreme genotypes were selected: rapidly dehydrated genotype Baghlani Nangarhar (NSF5) and slowly dehydrated genotype Saturn (NSF75); RT-qPCR expression analysis was performed for nine differentially expressed candidate genes, and their expression profiles were verified via transcriptome sequencing. The sample RNA was the residual RNA at the time of sequencing. RT-qPCR quantification was performed on the two genotypes with three biological replicates. The results show that the real-time PCR expression patterns of the nine genes are consistent with the RNA-seq expression profiles (Figure 6). This indicates that the RNA-seq expression profile of this experiment is authentic and reliable.

The transcriptome results show that genes *LOC4334080*(*LOC_Os03g53340*), *LOC4332360*(*LOC_Os03g16020*), *LOC4332361*(*LOC_Os03g16030*), and *LOC9267997*(*LOC_Os04g01740*) were up-regulated in the seed transcriptome of both the rapidly dehydrated genotype and slowly dehydrated with highly significant differences. *LOC4332360*, *LOC4332361*, and *LOC9267997* were significantly more expressed in the seeds of the rapidly dehydrated genotype than in the seeds of the slowly dehydrated genotype. *LOC4332360*, *LOC4332361*, and *LOC9267997* were significantly more expressed in the seeds of the slowly dehydrated genotype than in the rapidly dehydrated genotype, while the gene *LOC4334080* was more expressed in the rapidly dehydrated genotype than in the slowly dehydrated genotype. The gene function annotation revealed that *LOC4334080*, *LOC4332360*, *LOC4332361*, and *LOC9267997* belonged to heat-stimulated proteins or heat-stimulated transcription factors (Table 2). Based on the experimental results, it was hypothesized that the slow dehydration genotype was more sensitive to external environmental heat responses and induced stronger effects than the rapid dehydration genotype. The RT-qPCR quantification results further verify the reliability of the experimental results.

The transcriptome results show that genes *LOC4331608*(*LOC_Os03g05290*), *LOC4330265*(*LOC_Os02g44870*), and *LOC4326935*(*LOC_Os01g50700*) were up-regulated in the transcriptome of both the rapid dehydration Baghlani Nangarhar and slow dehydration Saturn seeds, and the differences were highly significant. *LOC4326935* was expressed in the rapid dehydration Baghlani Nangarhar and slowly dehydrated Saturn, with highly significant differences between pre- and post-dehydration phases and a slightly higher expression in the rapidly dehydrated genotype than in the slowly dehydrated genotype. This was consistent with the real-time quantitative PCR expression pattern; however, the results show that the expression levels in the rapidly dehydrated Baghlani Nangarhar were significantly higher than those in the slowly dehydrated genotype. *LOC4331608* and *LOC4330265* are dehydration-related genes (Table 1), and their expression levels in the rapid dehydration Baghlani Nangarhar were significantly higher than those in the slow dehydration Saturn. This indicates that the sensitivity of the dehydration-related genes to water metabolism is stronger in the rapidly dehydrated genotype than in the slowly dehydrated genotype and is presumably one of the reasons for the difference in the dehydration rates between them. RT-qPCR quantification further verified the reliability of the expression results of these three genes.

The transcriptome results show that the expression levels of genes *LOC4330248*(*LOC_Os02g44630*) and *LOC4343122*(*LOC_Os07g26690*) were both up-regulated in the rapidly dehydrated Baghlani Nangarhar seeds with highly significant differences, and the real-time quantitative PCR results show that the relative expression levels of these differentially expressed genes are consistent with the transcriptome results. The expression levels of genes *LOC4330248* and *LOC4343122* increased in the slowly dehydrated Saturn seeds but did not achieve a significant difference level. The results of the real-time quantitative PCR show that the expression levels of these two genes presented a significant difference in the slow dehydration genotype. This indicates that the induced expression of *LOC4330248* and *LOC4343122* was affected by the characteristics of the genotype itself and may be related to genotype specificity.

## 3. Discussion

The dehydration of mature seeds is a complex process in which seed dehydration tolerance is reduced by the loss of water, its metabolic activity is reduced, and the seed embryo enters a quiescent state or is metabolically inactive. Usually, dehydration causes some damage to the seeds, and the accumulated sugars, proteins, lipids, and enzymes of the antioxidant system in the seeds counteract the damage caused by dehydration and mitigate the effects on seed embryo viability by maintaining protein (including enzyme) stability, promoting cytoplasmic vitrification to protect subcellular stability, scavenging free radicals and other toxins, and other physiological and biochemical processes. During dehydration, the main processes involved in the seed’s response are signal transduction, protein folding, sorting and degradation, amino acid metabolism, sugar metabolism, lipid metabolism, biosynthesis of other secondary metabolites, energy metabolism, environmental adaptation, growth and death, membrane transport, transcriptional regulation, translation, replication and repair, metabolism of terpenoids and polyketides, nucleotide metabolism, glycan biosynthesis and metabolism, and aging and other metabolic pathways.

This experimental study showed that the differentially expressed gene *Os06g0159900* [Log2FC: 1.03], which was significantly expressed only in the slow dehydration genotype, was up-regulated and *LOC9266706* [Log2FC: −8.25] was down-regulated. *Os06g0722450* is an expression protein with an unknown function; it was not expressed during the initial state of the rapid dehydration genotype and was activated and up-regulated during the dehydration process. Its target mRNA is *novel-osa-miR194-3p* [23], which is presumed to be a new gene related to the dehydration rate. *Os06g0513943* was up-regulated in the rapid dehydration genotype; its main function is to maintain the integrity of the membrane, reduce physical damage to the seeds, and perform a self-defense role [24]. *LOC112938716* was down-regulated in the rapid dehydration genotype, indicating that its expression was inhibited during dehydration. *Os06g0159900* belongs to the U-box domain protein [25,26,27], which was highly up-regulated in the slow dehydration genotype. Its molecular function includes ubiquitin protein transferase and kinase activities, which can regulate the internal and external pressure balance levels of seed cells. *LOC9266706* is an expression protein with an unknown function; it was down-regulated by a large amount in the slow dehydration genotype and was expressed in the initial state. Following dehydration and drying treatments, the expression level was close to zero. It was hypothesized that the expression level was inhibited during the seed dehydration process; therefore, the expression level was extremely low. However, the gene was up-regulated in the rapid dehydration genotype Baghlani Nangarhar and was predicted to be a new gene, *novel-osa-miR116-5p*, through the collection of targeted miRNA information. In this study, two new genes related to dehydration were obtained, and their functions need to be further confirmed in future research.

The GO analysis of CNSF5 vs. NSF5 showed that 53 DEGs were enriched for a total of 112 entries. The most enriched entries of these differentially expressed genes were unfolded binding proteins and folded proteins, and the sites of action of these proteins were concentrated in the cytoplasm. CNSF75 vs. NSF75 GO analysis showed that 25 DEGs were enriched with a total of 65 entries, and the most enriched entries of these differentially expressed genes were unfolded binding and folded proteins; again, the cytosolic component was mainly enriched in the cytoplasm. These proteins play catalytic, binding, translocating, transcriptional, and other regulatory roles in adaptive responses to hydrogen peroxide, reactive oxygen species, high temperature, etc., in the adverse conditions to which they are exposed. In the comparison group CNSF5 vs. NSF5, we detected the gene for chitinase, a heat-shock protein, a result similar to that achieved by Chen [28]. Chen showed that the chitin receptor (*OsCERK1*) processed in the rice endoplasmic reticulum interacts with a heat-shock protein (Hsp90) and its accessory molecular chaperone Hop/Sti to form a complex that regulates the efficient transport of plasma membrane GTPase. The study of *maize* dehydration-rate-related genes that obtained specific genes at the transcriptional level of expression was closely associated with tissue senescence, the regulation of hormone levels, hormone-mediated signaling pathways, heat, reactive oxygen species, ethanol, and dehydration response [22], and the results are consistent with those obtained for the biological process of differentially expressed genes screened using GO enrichment analysis in this study. In a study of the functions of differential proteins obtained during the dehydration of wheat seed maturation, the main functions were considered to be stress/defense, storage proteins, energy metabolism, starch and sucrose metabolism, protein synthesis/folding/degradation, lipid metabolism, signal transduction, and transcription/translation [29]. These results are similar to those obtained in the present study. The molecular functions of the differentially expressed genes obtained in this study that were associated with the rate of dehydration were unfolded protein binding (GO:0051082), protein self-association (GO:0043621), and ADP binding (GO:0043531), while biologically important processes were protein folding (GO:0006457), response to reactive oxygen species (GO:0000302), and response to heat (GO:0009408). It was also hypothesized that heat-shock protein-related genes in the metabolic pathway (*LOC_Os03g16020*, *LOC_Os04g01740*, etc.), and the chitinase 2-like gene (*LOC_Os05g33130*), play an important regulatory role in the formation and efficient transport of plasma membrane GTPases.

The KEGG analysis showed that the most significantly enriched metabolic pathway for CNSF5 vs. NSF5 was the protein-processing function in the endoplasmic reticulum, with six differentially expressed genes involved in up-regulating the expression levels, mainly the small-molecule heat-shock proteins HSP16.6 (*LOC_Os01g04340*), HSP17.7 (*LOC_Os03g16040*), HSP17.4 (*LOC_Os03g16020*), and HSP18.1 (*LOC_Os03g16030*), the large-molecule heat-shock protein 82 (*LOC_Os04g01740*), and the heat-shock transcription factor HSF11 (*LOC_Os03g53340*) (Appendix A). The most significantly enriched metabolic pathway of CNSF75 vs. NSF75 was the protein-processing activity occurring in the endoplasmic reticulum, with eight differentially expressed genes involved in up-regulated expression and one in down-regulated expression levels. The differentially expressed genes involved in up-regulated expression levels were mainly HSP16.9 (*LOC_Os01g04360*), HSP17.8 (*LOC_Os02g48140*), HSP17.4 (*LOC_Os03g16020*), HSP18.1 (*LOC_Os03g16030*), HSP23.2 (*LOC_Os04g36750*), HSP16.0 (*LOC_Os06g14240*), HSP21.9 (*LOC_Os11g13980*), and heat-shock protein 82 (*LOC_Os04g01740*), and the gene involved in the down-regulated expression level was HSP81-3 (*LOC_Os09g30418*) (Appendix A). This is consistent with the results for small-molecule heat-shock proteins obtained in a study of the dehydration tolerance of Fraxinus mandshurica seeds by Liu [30]. Liu studied and obtained seven proteins related to dehydration tolerance levels: one up-regulated and six down-regulated proteins. The main biological functions of these seven proteins were related to energy metabolism, stress defense, and transcriptional regulation factors. The differentially expressed genes identified from the transcriptome analysis of Fraxinus mandshurica seeds were mainly involved in biological processes, such as protein folding, hydrogen peroxide reaction, heat adaptation, drying reaction, salt stress reaction, and hypertonic reaction. Some studies suggest that small-molecule heat-shock proteins can repair and degrade damaged proteins, stabilize polypeptide chains, maintain normal protein activity, and help improve the resistance of seeds to adverse conditions, thereby improving storage capacity outcomes related to seed lifespan results [31]. The research results (Zhang et al. [32]) suggest that there is a positive correlation between heat-shock proteins and the vitality of rice seeds [32]. The higher the content of heat-shock proteins in seeds with greater dry resistance, the stronger their seed vitality and the longer their storage time [32]. It is believed that the level of seed vigor is affected by the synthesis ability of heat-shock proteins in seeds, and the synthesis of a high number of heat-shock proteins could improve seed vigor [33]. It is believed that the ability of mature seeds to tolerate large amounts of water loss is related to the function of sHSPs, as sHSPs can protect cell components and avoid damage caused by water loss [34]. It is also believed that the main role of small-molecule heat-excited proteins is to maintain chaperone-folding activity and the folded or unfolded states to maintain limited intracellular water in dehydrated and desiccated environments, and to enhance seed-resistance outcomes [35]. Kermode concluded that normal (Normal seed refers to the ability to survive for decades, hundreds of years, or even thousands of years under low temperature and low humidity conditions.)seeds have greater dehydration tolerance under slow dehydration conditions, probably because slow dehydration induces the synthesis of protective substances associated with dehydration tolerance, the most important of which are heat-stable proteins, including LEA(late embriogenesis abundant proteins) proteins [36]. The heat-stimulated protein HSP maintains intracellular protein stability under normal conditions and stressful environments and helps proteins to complete proper folding, translation, and aggregation behaviors, which is important for enhancing drought tolerance levels in plants [37]. It has been suggested in previous research that HSP promotes cytoskeleton formation, protects cellular structural stability, and mitigates damage to membranes and cellular dehydration, thereby reducing overall cellular damage [38,39]. It has also been suggested in the literature that OsHSP18.2 is highly expressed during the seed maturation stage and acts as a molecular chaperone to protect the cell by reducing ROS accumulation levels, achieving enhanced seed viability and storage tolerance results [40]. In summary, heat-shock proteins and small-molecule heat-shock proteins can enhance the stability of the internal protein structure of seeds, improve seed vitality and storage life, and enhance the seed’s defense and resistance capabilities. Several types of heat-shock proteins (HSP16.9 (*LOC_Os01g04360*), HSP17.8 (*LOC_Os02g48140*), HSP17.4 (*LOC_Os03g16020*), HSP18.1 (*LOC_Os03g16030*), HSP23.2 (*LOC_Os04g36750*), HSP16.0 (*LOC_Os06g14240*), HSP21.9 (*LOC_Os11g13980*), heat-shock protein 82 (*LOC_Os04g01740*), and HSP81-3 (*LOC_Os09g30418*)) obtained in this study are presumed to have the effects of maintaining cell structure stability, heat resistance, and compression resistance during seed dehydration and drying treatments.

A pathway that was significantly enriched in KEGG in this study was oxidative phosphorylation metabolism, which is a coupled reaction in which eukaryotes release energy during material processing and oxidation processes occurring in the inner mitochondrial membrane and supply ADP through the respiratory chain to synthesize ATP with inorganic phosphate. *LOC107279585* (ATP synthetase subunit 9) was significantly up-regulated between comparison group CNSF5 vs. NSF5 and CNSF75 vs. NSF75. The difference in the results was due to the expression level of ATP synthetase subunit 9 in the rapidly dehydrated genotype being significantly higher than that in the slowly dehydrated Saturn, suggesting that the rapidly dehydrated genotype may maintain a high ATP synthesis and metabolic level, and provide sufficient energy for rapid dehydration. If ATP synthesis is blocked, this inhibits normal metabolic processes to some extent, which can lead to increased levels of reactive oxygen species and membrane lipid peroxidation, causing damage to seed viability [41]. The key differentially expressed genes and the most important metabolic pathways identified in this study for the rapidly and slowly dehydrated genotypes were protein-processing activity in the endoplasmic reticulum and oxidative phosphorylation metabolism, which are presumed to have important regulatory roles in stress/defense, energy metabolism, protein synthesis/folding, and signal transduction during the dehydration and drying processes of mature seeds.

## 4. Materials and Methods

### 4.1. Material Handling

The experimental materials were rice germplasm materials provided by the laboratory of Mr. Wang Guoliang from Hunan Agricultural University (Changsha, China). The rapidly dehydrated Baghlani Nangarhar (No. 19NSF5) and slowly dehydrated Saturn (No. 19NSF75) were selected based on the preliminary physiological maturity rice seed dehydration rate test (in which the dry matter weight in grains reached the maximum possible value). They were planted in Sanya, Hainan Province, South China, on 24 December 2020. Sowing was performed in stages based on the different seed-growing stages, with 20 seedlings planted per material, and a tag marking the date at the beginning of the spike, until 21 days following the flowering stage, when the sampling process began. Sampling method: 5 spikes of uniform maturity were selected for each sample, yellow ripe grains were threshed at the top 1/3 of the seeds and mixed, and the seeds were then placed in parchment paper bags and subjected to rapid-dehydration conditions (constant temperature 45 ± 2 °C dryer). Their initial moisture-content level was recorded, and the samples were set up in three replicates. The control seeds (in the state at harvest) were rapidly packed into 2.0 mL Eppendorf tubes, immediately snap-frozen in liquid nitrogen, and then stored in the laboratory in an ultra-low-temperature refrigerator for later transcriptome sequencing. The treated seeds were rapidly dehydrated, and the moisture content was measured every 4 h until it stabilized at 13%; then, seeds were immediately snap-frozen in liquid nitrogen and stored in an ultra-low-temperature refrigerator at −80 °C.

### 4.2. Methods

#### 4.2.1. Seed RNA Extraction

An RNAprep pure Plant Kit (Tiangen, Beijing, China) was used to extract the seed RNA information. An Agilent 2100 Bioanalyzer (Agilent Technologies, Santa Clara, CA, USA) and NanoDrop ND-2000 spectrophotometer (NanoDrop Technology, Wilmington, DE, USA) were used to assess RNA concentration, integrity, and purity levels. We then analyzed all the RNA samples with OD260/OD280 ratios between 1.8 and 2.2, including a pooled RNA sample for transcriptome sequencing and RNA preparations from CNSF5, NSF5, CNSF75, and NSF75 samples for DEG sequencing. We selected cDNA libraries with RNA integrity number (RIN) values greater than 7.6 for quantitative real-time PCR.

#### 4.2.2. Transcriptome Sequencing and Data Assembly

Following the RNA extraction process, each sample was analyzed for its quality. Quality analysis, library construction, and sequencing were performed by UW Genetics Biotechnology Co. using SOAPnuke (v1.5.2) [42]. The sequencing data we obtained were then filtered by (1) removing the reads that contained sequencing adapters; (2) removing the reads with a low-quality base ratio greater than 20% (base quality less than or equal to 5); (3) removing the reads with an unknown base (‘N’ base) ratio of more than 5%, and then retrieving clean reads and storing them in FASTQ format. Clean reads were mapped to the reference genome using HISAT2 (v2.0.4) [43]. Bowtie2 (v2.2.5) [44] was then applied to align the clean reads to the reference coding gene set (reference genome version: GCF_001433935.1_IRGSP-1.0, https://biosys.bgi.com/assets/img/banner1.svg/on accessed on 10 August 2021), and then RSEM (v1.2.12) [45] was used to calculate the expression levels of the genes. A heat map was created using pheatmap (v1.0.8) [46] based on the gene expression levels in different samples.

#### 4.2.3. Functional Annotation, Classification, and Metabolic Pathway Analysis

(1)Analysis of differentially expressed genes (DEGs). A total of four comparison groups were constructed: CNSF5 vs. NSF5, CNSF5 vs. CNSF75, NSF5 vs. NSF75, and CNSF75 vs. NSF75. The results of the clean reads were compared with the reference genome and stored in binary files. Gene FPKM (FPKM = total exon fragments/mapped reads (Millions) × exon length (kb)) [47] was quantified using Cufflinks [48]. The number of reads of genes in the samples was obtained using HTSeq-count [49] (California Institute of Technology, Pasadena, CA, USA) software. The data were normalized using the software DESeq2 [50], the R package was used to estimate the size factor function for normalization, and the nbinom test function was used to calculate fold-change values and *p*-values for comparative differences to control the false discovery rate. DEGs were selected with *p*-values < 0.05. After corrections were performed, a rigorous threshold (Q-value 0.05) was utilized based on the method of Bonferroni [51] to correct the significance levels of terms and pathways.(2)Enrichment analysis of DEGs via GO and KEGG. GO and KEGG enrichment analyses were performed on the screened differentially expressed genes using GO (http://www.geneontology.org accessed on 1 May 2023) [52] and KEGG (https://www.kegg.jp accessed on 1 May 2023) databases [53]. Term and pathway significance was assessed using the corrected Q-value < 0.05 [51].

#### 4.2.4. RT-qPCR Validation of Differentially Expressed Genes

We used Premier Software to design the appropriate RT-qPCR primers and used a cDNA synthesis kit from TransGen Biotech to reverse-transcribe the RNA into cDNA (see cDNA synthesis instructions). The reagents used in this study to perform quantitative PCR analyses were Takara’s quantitative PCR kits, and the total reaction system was 50 μL, including 1 μL of reverse-transcription product, 1 μL of gene-specific primers, 25 μL of 2× Trans TaqTM HiFi PCR SuperMix II, and 3 μL of ddH_2_O. Amplification was performed using an ABIPRISM7500 (Applied Biosystems, Waltham, MA, USA) model real-time fluorescent quantitative PCR instrument. The genes were assessed quantitatively using the 2^−∆∆Ct^ algorithm [54], and significance analysis (one-way ANOVA and multiple comparisons, *p* < 0.05) was performed using SPSS 13.0 to correct for differential gene expressions according to the internal reference gene actin. Three replicates were set up for each PCR. Primer-specific information is provided in Table 3.

## 5. Conclusions

Transcriptomic sequencing analysis performed on two varieties of mature rice seeds (extremely-rapid-dehydration Baghlani Nangarhar and slow-dehydration Saturn) showed that the number of up-regulated expression difference genes of the two genotypes following dehydration treatment was significantly greater than that of the initial material, and the number of differentially expressed genes of the rapid-dehydration genotype was significantly greater than that of the slow-dehydration genotype. Through the GO enrichment analysis, it was observed that the proteins encoded by these differentially expressed genes related mainly to unfolded protein binding, heat-shock protein binding, protein self-association, ADP binding, protein heterodimer active-protein heterodimerization activity, UDP-glycosyltransferase activity, and ligand-gated ion channel activity. It might be surmised that these proteins are closely related to the seed dehydration rate. The results of this study provide a valuable reference for further research on the genes and metabolic pathways related to the dehydration rate of mature rice seeds and provide theoretical guidance for the selection and breeding of new rice germplasm that can be rapidly dehydrated at the mature stage. Subsequent expression and functional analyses of the mutants of these genotype materials need to be carried out.

## Figures and Tables

**Figure 1 ijms-24-11527-f001:**
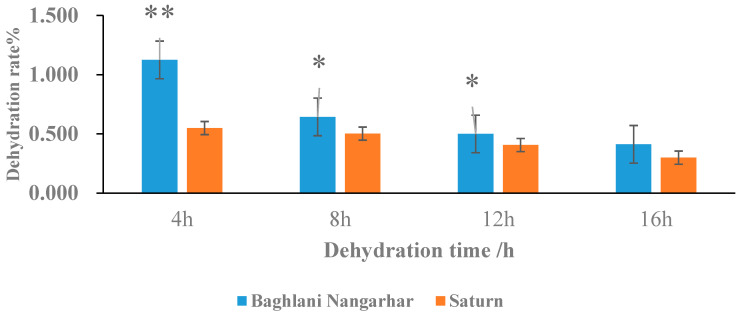
Variation in dehydration rate levels of rice seeds exposed to rapid dehydration conditions. *, significant at *p* < 0.05; **, significant at *p* < 0.01.

**Figure 2 ijms-24-11527-f002:**
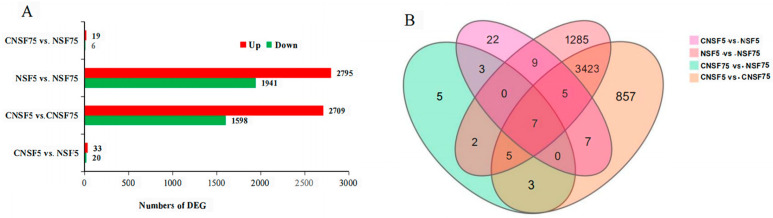
Overview of RNA-seq in the 4 comparison groups of DEGs: (**A**): up- and down-regulation of DEGs in four comparison groups. Red and green bars represent significantly up- and down-regulated genes, respectively; (**B**): DEG distribution in Venn diagram of the 4 comparison groups.

**Figure 3 ijms-24-11527-f003:**
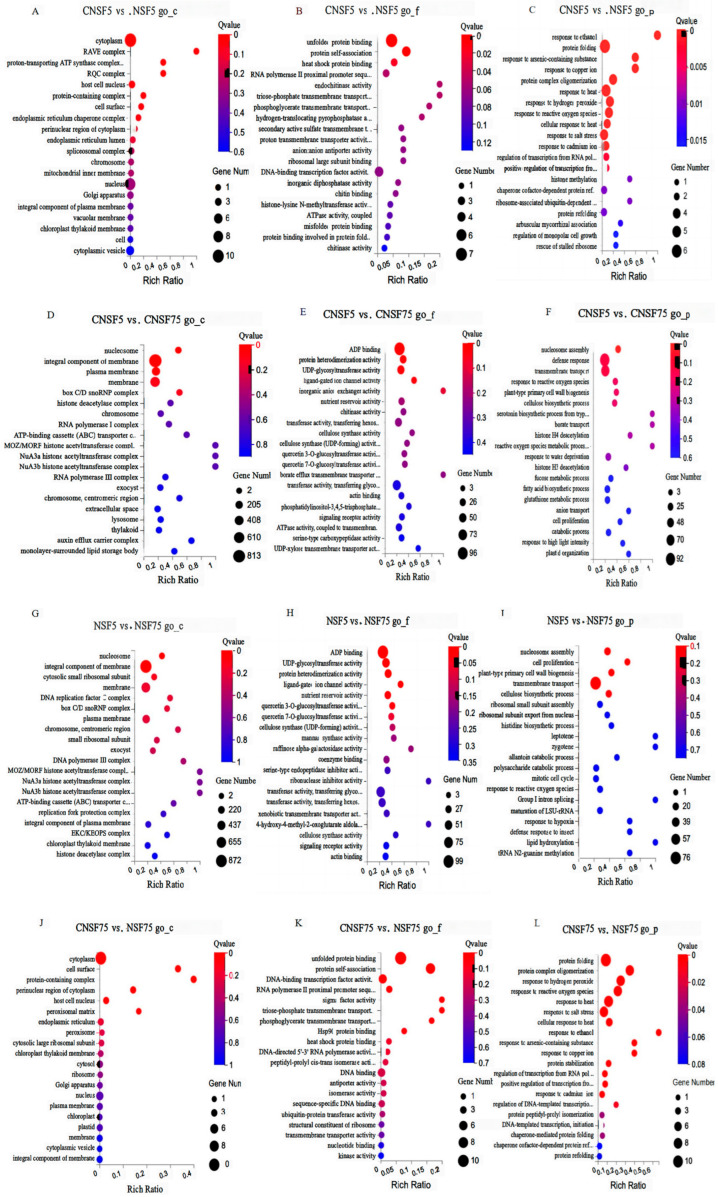
Overview of GO enrichment analysis of four comparison groups: DEGs (**A**–**C**) denote the cell components, molecular functions, and bioprocess GO enrichment of CNSF5 vs. NSF5 in the comparison group; (**D**–**F**) denote the cell components, molecular functions, and bioprocess GO enrichment of CNSF5 vs. CNSF75 in the comparison group; (**G**–**I**) denote the cell components, molecular functions, and bioprocess GO enrichment of NSF5 vs. NSF75 in the comparison group; (**J**–**L**) denote the cell components, molecular functions, and bioprocess GO enrichment of CNSF75 vs. NSF75 in the comparison group. GO terms are listed on the Y-axis, and rich factors are shown on the X-axis. A lower Q-value indicates a higher significance level and closer proximity to the red line. A larger circle represents more genes.

**Figure 4 ijms-24-11527-f004:**
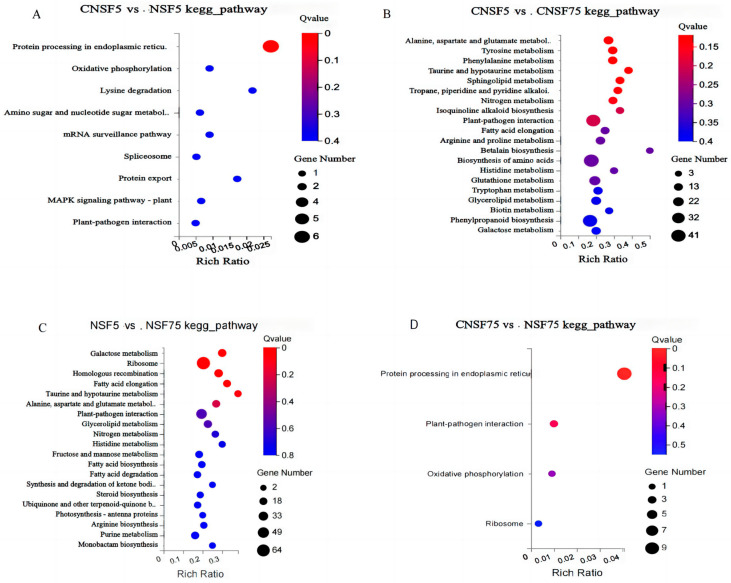
Analysis of differentially expressed genes via KEGG enrichment: (**A**–**D**) represent the most significant KEGG pathway enrichment maps of CNSF5 vs. NSF5, CNSF5 vs. CNSF75, NSF5 vs. NSF75, and CNSF75 vs. NSF75 in the comparison group. GO terms are listed on the Y-axis, and rich factors are shown on the X-axis. A lower Q-value indicates a higher significance level and closer proximity to the red line. A larger circle represents more genes.

**Figure 5 ijms-24-11527-f005:**
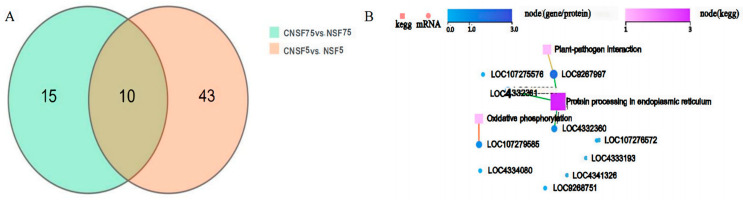
Overview of shared differentially expressed genes and gene network interactions. (**A**): Venn diagram of DEGs in groups CNSF75 vs. NSF75 and CNSF5 vs. NSF5; (**B**): interaction map of shared significantly differentially expressed gene networks in groups CNSF5 vs. NSF5 and CNSF75 vs. NSF75.

**Figure 6 ijms-24-11527-f006:**
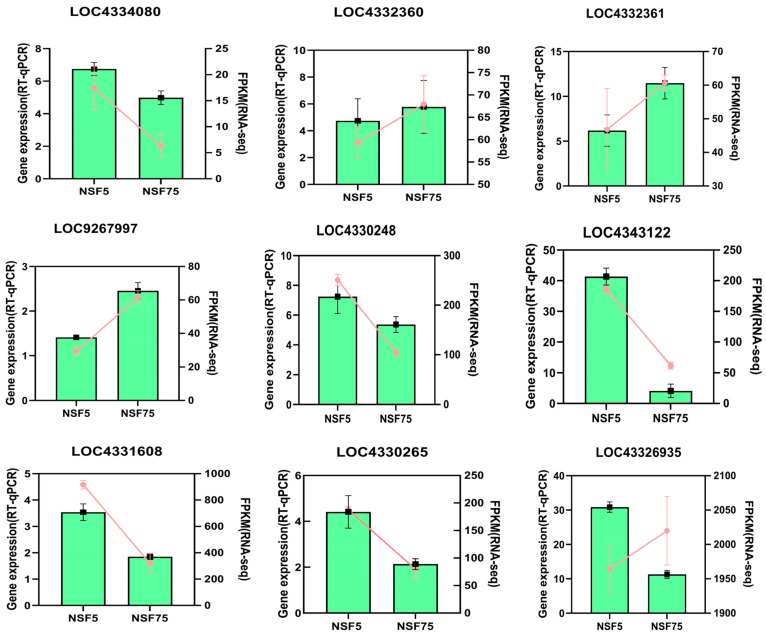
Validation of DEGs via RNA-seq and RT-qPCR methods. Note: histograms represent the results of the RT-qPCR assays, using the 2^−∆∆Ct^ algorithm, with the scale on the left ordinate of each graph. Orange lines represent the results of the FPKM analyses, with the scale on the right ordinate of each graph.

**Table 1 ijms-24-11527-t001:** Shared differentially expressed genes in CNSF75 vs. NSF75 and CNSF5 vs. NSF5.

NCBI Gene No.	MSU Gene No.	Variance Multiplier		Gene Annotation
		log2 (NSF5/CNSF5)	log2 (NSF75/ CNSF75)	
*LOC4341326*	*LOC_Os06g36930*	4.3997	4.3444	Putative heat-stress transcription factor A-6a
*LOC4334080*	*LOC_Os03g53340*	1.1557	1.7690	Similar to heat-stress transcription factor A-2a
*LOC9268751*		2.2410	1.7960	Ethylene-responsive transcription factor ABR1
*LOC107279585*		1.4887	1.1606	ATP synthase subunit 9, mitochondrial
*LOC4332361*	*LOC_Os03g16030*	1.9368	2.3599	Low-molecular-mass heat-shock protein Oshsp18.0
*LOC9267997*	*LOC_Os04g01740*	1.5389	1.7422	Similar to heat-shock protein 82
				Heat-shock protein 81-1 (HSP81-1) (Heat-shock protein 83)
				Similar to heat-shock protein 80
				Non-protein-coding transcript
*LOC4332360*	*LOC_Os03g16020*	1.3314	1.2284	Low-molecular-mass heat-shock protein Oshsp17.3

**Table 2 ijms-24-11527-t002:** Functional information for the nine candidate genes.

NCBI Gene No.	MSU No. or RAP No.	Function Comments	Molecular Function	Biological Process
*LOC4334080*	*LOC_Os03g53340*	Similar to heat-stress transcription factor A-2a	GO:0000978 RNA polymerase II proximal promoter sequence-specific DNA binding; GO:0003677 DNA binding; GO:0003700 DNA-binding transcription factor activity; GO:0043565 sequence-specific DNA binding	GO:0006355 regulation of transcription, DNA-templated; GO:0034605 cellular response to heat; GO:0043618 regulation of transcription from RNA polymerase II promoter in response to stress; GO:0061408 positive regulation of transcription from RNA polymerase II promoter in response to heat stress
*LOC4332360*	*LOC_Os03g16020*	17.4 kDa class I heat-shock protein-like	GO:0043621 protein self-association; GO:0051082 unfolded protein binding	GO:0000302 response to reactive oxygen species; GO:0006457 protein folding; GO:0009408 response to heat; GO:0009651 response to salt stress; GO:0042542 response to hydrogen peroxide
*LOC4332361*	*LOC_Os03g16030*	18.1 kDa class I heat-shock protein-like	GO:0043621 protein self-association; GO:0051082 unfolded protein binding	GO:0000302 response to reactive oxygen species; GO:0006457 protein folding; GO:0009408 response to heat; GO:0009651 response to salt stress; GO:0042542 response to hydrogen peroxide
*LOC9267997*	*LOC_Os04g01740*	Heat-shock protein 82	GO:0005524 ATP binding; GO:0051082 unfolded protein binding;	GO:0006457 protein folding; GO:0034605 cellular response to heat; GO:0050821 protein stabilization
*LOC4331608*	*LOC_Os03g05290*	Probable aquaporin *TIP1-1*	GO:0015250 water channel activity; GO:0015267 channel activity	GO:0006833 water transport; GO:0055085 transmembrane transport
*LOC4330265*	*LOC_Os02g44870*	Dehydrin *DHN1*-like	Unknown	GO:0006950 response to stress; GO:0009414 response to water deprivation; GO:0009415 response to water; GO:0009631 cold acclimation; GO:0009737 response to abscisic acid
*LOC4326935*	*LOC_Os01g50700*	Dehydrin *Rab25*-like	Unknown	GO:0009414 response to water deprivation; GO:0009415 response to water; GO:0009631 cold acclimation; GO:0009737 response to abscisic acid
*LOC4330248*	*LOC_Os02g44630*	Aquaporin *PIP1-1*-like	GO:0015250 water channel activity; GO:0015267 channel activity	GO:0006833 water transport; GO:0009414 response to water deprivation; GO:0055085 transmembrane transport
*LOC4343122*	*LOC_Os07g26690*	Probable aquaporin *PIP2-1*	GO:0005215 transporter activity; GO:0015250 water channel activity; GO:0015267 channel activity	GO:0006810 transport; GO:0006833 water transport; GO:0055085 transmembrane transport

**Table 3 ijms-24-11527-t003:** Primers used for the RT-qPCR assay in this study.

Serial Number	NCBI Login Number	Candidate Genes	Primer Sequences
1	*LOC4343122*	*LOC_Os07g26690*	F: TGTTTAGCCTGTACTCCCATTT
			R: ACGGAGGGAGTATATTCCAGAT
2	*LOC4332360*	*LOC_Os03g16020*	F: GCATTGGGCTAATCTAAAACGA
			R: GCACACCAAAAACACCAGTAAT
3	*LOC4332361*	*LOC_Os03g16030*	F: GGTTACCGGCTAGTAAGAAACT
			R: TACTGCAATTGATCACAAACCG
4	*LOC4334080*	*LOC_Os03g53340*	F: CTACGAAGGTCGATCCGGATAG
			R: CTTGATCGTCTTCAGGAGCTC
5	*LOC9267997*	*LOC_Os04g01740*	F: GGAGGAGGTGGACTGAATTAAA
			R: ACTTTCTCAACGATGGCTTAGA
6	*LOC4330248*	*LOC_Os02g44630*	F: CATTCAAGAGCAGGTCTTAAGC
			R: AGTTGTTCAGGGTTCAGATAGG
7	*LOC4331608*	*LOC_Os03g05290*	F: GAGTCCCAGTGGGTGTACT
			R: GAGATGAAGAGGACCTCGTAGA
8	*LOC4330265*	*LOC_Os02g44870*	F: GAGAAGATCGAGGGTGATCAC
			R: GCTTCTCCTTGATCTTGTCGAG
9	*LOC4326935*	*LOC_Os01g50700*	F: CAGTCGTGTTTCAGTTCGTTAA
			R: GGATACACCGTACATGCATAGA

## Data Availability

Availability statements are provided in the section “MDPI Research Data Policies” at https://www.mdpi.com/journal/ijms/instructions (accessed on 11 July 2023).

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
