# Peer review of "Transcriptomic Analysis of the Dehydration Rate of Mature Rice (Oryza sativa) Seeds"

_ijms, 2023, doi:10.3390/ijms241411527_

Round 1

Reviewer 1 Report (Previous Reviewer 1)

The paper looks better now, however, it is still not easy to read it. Authors need to go through text carefully and made at least grammar/style corrections.

Below are some points:

Lines 10–11: two times dehydration rate; I am not sure you provide, you rather can provide theoretical basis.

Line 15: “ Fast dehydration experiment on these two types of seeds was conducted.”?? Two types of seeds or two genotypes?  You can not mention this term here because reader did nit kniow what is fast dehydration. Moreover, I think it is better write not fast, but rapid.

Line 19: “  qRT-PCR” = qPCR.

Line 20: “The results showed that in Baghlani Nangarhar, 53 differentially expressed

genes (DEGs)  were screened”??? Completely wrong grammar here. Please, re-write.

Line 22: what is “Similarly”???

Lines 17 and 28: two times They, but seneteces are in different tenses. Why?

Lines 187-188: physiological maturity and increasing maturity: can you be more precise and explain what do you mean as physiological maturity?

Figure 1 is not clear. Please, provide short introduction what do you mean as dehydration time , what is % etc.

Lines 232- 233: maybe it will be better to use a table instead of etxt to make link more clear. And, of course, description of the groups (NSF5 etc) should be done here, not on the lines 753-756. How reader can know what do you mean as NSF5 etc?

 Line 245: “rice” is redudndant here. Please, check whole text, replace material with genotype, avoid too long sentences.

Line 252: species? Which species do you mean? Maybe genotype?

Line 360: why capital? “Four”?

Line 389: “amino acid sugar” ?

Lines 443- 447: too long and complicated sentence.

Lines 519 - 521: “Based on the results of the previous screening of 165 germplasm materials for

dehydration rate phenotypes, two extreme materials were selected: the fast -dehydrating

material Baghlani Nangarhar (NSF5) and the slow -dehydrating material Saturn (NSF75),” ?? This message not new, why did you repeat it?

Line 619 provide the same information as line 264.

Line 682: “ The research results” ??

Line 741: “Sowing was carried out in stages based on the different seed-growing

stages” ??? Not clear what do you mean.

Line 742: “ 20 plants planted for each material”? = “each line” (or genotype).

Line 751: how did you measure moisture contents?

Lines 753-756: the name of the group should be explained somehow in lines 197-198, ea. Before first mentioning . How reader can now these nomenclature if you mentioned it only on the lines 753?

Line 824: “the number of up-regulated expression difference genes” ??

material = genotype.

style/grammar! Many repetitive words in single senteces...

Author Response

Reviewer 2 Report (Previous Reviewer 2)

The goal of this manuscript is to present and discuss the significant role of changes in the transcriptome of mature Oryza sativa seeds with the different dehydration rate. For this purpose the authors had one approache by analyzing transcriptome profiles using RNA-seq data.
It is a resubmitted manuscript and this paper by Liu et al. has clearly benefited from the revision, as advised by reviewers. All parts of the manuscript is interesting and clearly summarize new data valuable for the research community. I still recommend revision of the English language by a native speaker or a commercial entity before publication.

GENERAL COMMENTS:
TITLE
The paper title is well stated, it is informative and concise.

ABSTRACT, INTRODUCTION
Abstract is very well written with the key findings of the study. Introduction is concise, focused and informative.

MATERIAL AND METHODS
Material and research methods are presented appropriately and clearly. Experimental setup and the description in the methods section are well structured, and the statistical analysis is done alright.

RESULTS
The results obtained in this study are interesting. Results presented concise and correct.

DISCUSSION AND CONCLUSIONS
In general, the discussion of results and conclutions are correct and sufficient.

LITERATURE
The items of literature included in the paper are rather sufficient and adequate to the subject of the paper.

I recommend revision of the English language by a native speaker or a commercial entity.

Round 2

Reviewer 1 Report (Previous Reviewer 1)

Line28: grammar!

Line 126: the concentration level ??

Lines 122 -124: please, provide detailed abbreviations of CNSF5 etc here, not in the M&M. How I can understand results without description?

Line 147: „The abovementioned“ – space.

minor corrections

Author Response

请参阅附件。

This manuscript is a resubmission of an earlier submission. The following is a list of the peer review reports and author responses from that submission.

Round 1

Reviewer 1 Report

The authors performed transcriptomic análisis of dehydration rate in rice seedlings.

The topic is interesting, however, in the present form it is impossibhle to read this paper.

Please, re-write text in a proper way and resubmit again.

Almost each sentence require clarification!

Below are only some some comments:

Lines 7-9: there rae 3 times rice and two times rice seeds. Since you stdied only rice, once time is enough.

Lines 9-11: it is not clear how did you select the cultivar: have you checked all cultivars from all 82 countries ori t was anotre algorithm?

Line 13: fluorescence PCR?  Do you mean qPCR?

Lines 13-17: differently expressed to compare with what?

Lines 17- 22: too long and complicated sentence with several messages in one. Please, Split to 2 or 3 with clear message in each. Hydrogen peroxide is a reactive oxygen species, by the way..

Line 24: differentialy expressed proteins??  You study mRNA, not protein.

Line 25: you did not obtained, you only characterised.

Line 32: normal sex seeds?

Line 36: “at maturity in production”??

Lines 37-41: very long with several messages. Please, Split!

“btained”??

Line 42: “how dehydration genes stress dehydration stress signals” what do you mean here?

Line 43: “how gene co-expression network mechanisms”???

Lines 45-48: long unclear sentence.

Lines 75-77: nuclear points.

in many casees spaces between senteces are missing-

The text is required re-writing to be readable.

Author Response

Dear Reviewer,

Thank you and best regards.

Yours sincerely,

Zhongqi Liu

Reviewer 2 Report

The goal of this manuscript is to present and discuss the sisgnificant role of changes in the transcriptome of mature Oryza sativa seeds with the different dehydration rate. For this purpose the authors had one approache by analyzing transcriptome profiles using RNA-seq data.
All parts of the manuscript is interesting and clearly summarize new data valuable for the research community. The manuscript is elaborated on an interesting topic, but requires modifications and additions before publication. The manuscript has many grammar and syntax errors that must be corrected prior to publication- please check the whole manuscript, this will help readers to read and understand your paper. I have the following comments and suggestions for the authors to improve the quality of the manuscript.

GENERAL COMMENTS:
TITLE
The paper title is well stated, it is informative and concise.
Line 3: Correct 'rice((Oryza sativa)) seeds' to 'rice (Oryza sativa) seeds'.

ABSTRACT, INTRODUCTION
Abstract is very well written with the key findings of the study. Introduction is concise, focused and informative.

Line 21: 'hydrogen peroxide' should be deleted.
Line 37: Correct 'costs.Some' to 'costs. Some'. The lack of spaces between words occurs many times throughout the manuscript- Authors should check the text before submitting it for review.
Line 39: Correct 'Trichoderma seeds' to 'Trichoderma'.
Line 42: Correct 'dehydration tolerance tolerance' to 'dehydration tolerance'. Word repetitions occur in many places in the manuscript - please check carefully.

MATERIAL AND METHODS
Material and research methods are presented appropriately but not clearly. Experimental setup and the description in the methods section are well structured, and the statistical analysis is done alright. In spite of that I have a few objections against its present form. These are listed below:

- Where is the RNA-seq data deposited?
- Supplementary data: Schedule 1- The authors used RNA libraries of poor purity, as indicated by the value of OD260/230 - please explain.
- What formula was used for calculation of FPKM values?

RESULTS
The results obtained in this study are interesting. Results presented concise, but not correctly yet.

DISCUSSION
In general, the discussion of results is correct and sufficient.

CONCLUSION
You have to re-write the conclusion because it is seemed as a repeat for the results without any key message. You have to wrap up your ideas and leave the reader with a strong final impression. I suggest to writing two other words on the aspect concerning the aspects where the future studies must be oriented.

LITERATURE
The items of literature included in the paper are rather sufficient and adequate to the subject of the paper. Please verify the correctness of the literature.

The manuscript has many grammar and syntax errors that must be corrected prior to publication. I recommend revision of the English language by a native speaker or a commercial entity.

Author Response

(The authors gave the same response as above.)

Round 2

Reviewer 1 Report

The text is significantly better, however some extra efforts may require.

Line 69: fluorescence is redundant.

 Lines 71- 75: “In the Baghlani Nangarhar, 53 DEGs were screened, of which 33 were up-regulated and 20 were down-regulated; in the slow dehydrating material and in Saturn, 25 DEGs were screened, of which 19 were up-regulated and 6 were down-regulated.” – you did not explain what do you mean. You need to clearly mentioned if you compare DEG in these two lines.

Please, add well formulated conclusions before M&M. Please, consider that transcriptomic analysis is rather a mirror of the mechanism, but not a hammer. Try to build some even speculative model what can be the “hammer”. Maybe hormone metabolism as ANA/auxin rate etc, what can be detected on the level of some proteins activity, not by transcriptome analysis.

minor corrections

Reviewer 2 Report

This paper by Liu and Zhang has clearly benefited from the revision, as advised by reviewers.

Extensive editing of English language is still required.

Round 3

Reviewer 1 Report

The text is better now, but unfrotunately, new part is not very clearly written.

For example:

"Line 487: what is “differential gene” ??

Line 503: “the expression amount was 0” ??

Line 533: “It is indicated that heat shock protein, chitinase, reactive oxygen species, and protein folding enriched in dehydration rate-related genes…” – please, avoid such a mixing: here you mix gene expression, with ROS, etc.  Please, re-formulate.

minor polishing still require
